# The Emerging Role of the Gut Virome in Health and Inflammatory Bowel Disease: Challenges, Covariates and a Viral Imbalance

**DOI:** 10.3390/v15010173

**Published:** 2023-01-06

**Authors:** Daan Jansen, Jelle Matthijnssens

**Affiliations:** Laboratory of Viral Metagenomics, Rega Institute, Department of Microbiology, Immunology and Transplantation, University of Leuven, B-3000 Leuven, Belgium

**Keywords:** IBD, virome, virota, bacteriophages, eukaryotic viruses, metagenomics

## Abstract

Virome research is a rapidly growing area in the microbiome field that is increasingly associated with human diseases, such as inflammatory bowel disease (IBD). Although substantial progress has been made, major methodological challenges limit our understanding of the virota. In this review, we describe challenges that must be considered to accurately report the virome composition and the current knowledge on the virome in health and IBD. First, the description of the virome shows strong methodological biases related to wetlab (e.g., VLP enrichment) and bioinformatics approaches (viral identification and classification). Second, IBD patients show consistent viral imbalances characterized by a high relative abundance of phages belonging to the *Caudovirales* and a low relative abundance of phages belonging to the *Microviridae*. Simultaneously, a sporadic contraction of CrAss-like phages and a potential expansion of the lysogenic potential of the intestinal virome are observed. Finally, despite numerous studies that have conducted diversity analysis, it is difficult to draw firm conclusions due to methodological biases. Overall, we present the many methodological and environmental factors that influence the virome, its current consensus in health and IBD, and a contributing hypothesis called the “positive inflammatory feedback loop” that may play a role in the pathophysiology of IBD.

## 1. An Introduction to the Virota as an Essential Component of the Human Microbiota

The human gut harbors a complex ecosystem consisting of viruses, archaea, fungi, bacteria, and protozoa. The collective of microorganisms is referred to as the “gut microbiota” and provides a considerable benefit to the human host [1]. Specifically, the gut microbiota promotes gut barrier integrity, nutrient production (e.g., vitamins and short-chain fatty acids (SCFA)), immune modulation, drug metabolization, and protection against invading pathogens [1]. Early attempts to characterize the gut microbiota involved culturing microbes, which showed limited success, as only a small proportion of gut microbes can be cultured [2]. Subsequently, the arrival of next-generation sequencing (NGS) technologies in the early 2000s allowed researchers to investigate the gut microbiota in all its complexity [3]. The microbiome field was born out of this technological success aiming to study the collective of microbial genomes, which paved the way for the advent of virome research as a subcategory. Soon afterwards, both the microbiome and virome fields experienced an explosion of scientific research, which is reflected by an average increase in year-over-year publications between 2011 and 2021, amounting to 33.4% and 28.1%, respectively (Figure 1). As a result, the International Committee on the Taxonomy of Viruses (ICTV) reported an unprecedented increase in the number of viral species between 2011 and 2021 (Figure 2). Even though considerable progress has been made, the virome is still an underrepresented subcategory (3.54%) of the total microbiome, as measured by research output (Figure 1). Regardless of the underrepresentation, there are frequent reports indicating that a disrupted gut virota, referred to as “viral gut imbalance”, is associated with various diseases [4]. These include type 1 diabetes, type 2 diabetes, obesity, cystic fibrosis, acquired immunodeficiency syndrome, graft-versus-host disease, colorectal cancer, and arguably the most well-studied disease related to the gut virome, inflammatory bowel disease (IBD) [5,6,7,8,9,10,11,12]. IBD is a group of chronic diseases with unknown etiology, and it affects millions of people worldwide [13]. It is characterized by episodic inflammation of the gastrointestinal tract and is typically divided into ulcerative colitis (UC) and Crohn’s disease (CD) [14]. UC is characterized by continuous inflammation of the colonic mucosa, whereas CD is characterized by patchy transmural inflammation that can affect any single layer of the gastrointestinal tract [14]. Despite the recognition of a viral gut imbalance in IBD, the role of the gut virome in disease pathophysiology remains enigmatic.

This review summarizes our current understanding of the gut virota in all its complexity with a specific focus on IBD patients. Here, we discuss current challenges in describing the virome composition, the environmental factors that shape it, and the consensus on the healthy (‘balance’) and diseased state (‘imbalance’) of the intestinal virota. Ultimately, we aim to provide crucial insights into the gut virota and its potential impact on the pathophysiology of IBD patients.

## 2. The Gut Virota as a Reproducing, Communicating, and Functional Unit

The human gut virota includes prokaryotic and eukaryotic viruses capable of infecting all three domains of life (i.e., eukaryota, bacteria, and archaea) and constitute single- or double-stranded RNA or DNA genomes [15]. Viruses are the most abundant biological entities in the world, even though their abundance varies immensely between environments [16,17]. For example, oceanic viruses outnumber bacterial cells by an order of magnitude, whilst Reyes and colleagues reported that the number of intestinal viruses is similar to that of intestinal bacteria [16,17]. Regardless, bacterial viruses enter their cellular host and exploit their infrastructure to initiate their own reproductive process. This process is broadly characterized by two lifecycles (e.g., lytic and lysogenic) in bacterial viral reproduction. The lytic cycle involves viral attachment, entrance, replication, and production of virions, which is (often) followed by lysing the cellular host [18]. In contrast, the lysogenic cycle involves the integration of phage genomic DNA within the cellular genome, or the formation of extrachromosomal plasmids within the cytoplasm [18]. In the lysogenic lifecycle, viruses can exist in a dormant state (i.e., as prophages), in which the genetic material will be transmitted to cellular progeny with the event of cell division [19]. Depending on the specific phage, their induction occurs either spontaneously at a low rate, or is triggered by external stressors, thereby activating the DNA damage response or SOS response [19,20]. The inducing agents in the human gut include changes in pH, nutrients, oxygen levels, antibiotics, hydrogen peroxide, reactive oxygen species (ROS), and other DNA-damaging agents [19,21,22]. In addition, Erez and colleagues observed a phage-specific communication system, called the “arbitrium system”, that allows the phage to sense its concentration in the environment, and decide whether to enter the lytic or lysogenic state [18]. With this system, phages lysogenize the host when there is a high concentration of the phage-encoded communication peptide (i.e., arbitrium) and will lyse the host based on a low concentration of arbitrium [18]. Next, prophages are a major genetic resource present in approximately 40–50% of microbes, representing as much as 20% of their genomes [19,23]. They provide a functional reservoir of phage-encoded genes available to the bacterial host [24]. The genetic reservoir can include functions promoting virulence and biofilm formation, providing metabolic and antibiotic-resistance genes, and improving stress tolerance and immunity [25,26,27,28,29,30,31]. Subsequently, the newly acquired bacterial functions can be beneficial (e.g., stimulating immunity) or detrimental (e.g., virulence factors) to the human host. One of the mechanisms providing bacterial immunity is superinfection exclusion. This way, prophages can provide an antiviral state by protecting bacterial cells from a secondary infection with a closely related phage [25]. Furthermore, prophages can spread virulence factors that convert gut commensals to gut pathogens [28,32]. For example, cholera-toxin encoding phages belonging to the *Inoviridae* family can integrate into the bacterial host, deliver cholera-toxin genes, and create pathogenic bacteria capable of spreading disease [32,33]. Interestingly, inoviruses can cause chronic infections that allow continuous virion production without killing the host [34].

## 3. Wetlab Challenges in Describing the Virome Composition

The identification and characterization of the human gut virota is based on a specific combination of wetlab and drylab (bioinformatic) methods. Differences in the employed methods will have an impact on the observed virome composition [35,36,37,38,39]. Arguably, two of the most important wetlab factors are the viral enrichment, sometimes called virus-like particle (VLP) purification, and the random DNA amplification. The enrichment of the viral genomic material is often necessary due to the smaller genome sizes of viruses compared to other microbes (e.g., bacteria), thereby representing only a small percentage of the genomic material within the human gut microbiome [35,40]. In addition, the concentration procedures can also enrich VLPs derived from prophages, which in the absence of external triggers, are oftentimes still expressed in low amounts [20]. The most common VLP purification steps are centrifugation, filtration, and nuclease treatment. Even though a number of virome studies employ VLP purification steps, the precise implementation can still differ greatly between studies, mainly with regard to centrifugation speed and filter pore sizes, thereby affecting the observed virome composition [17,41,42,43,44]. Next, a random amplification step (with or without reverse transcribing RNA viruses) is often used when the amounts of genetic material are too low for further processing. A known example is multiple displacement amplification (MDA), which is believed to introduce an amplification bias towards ssDNA circular viral genomes [45]. Finally, regarding the sequencing process, the choice of platform and the sequencing depth also influence the virome composition [46,47,48]. For example, a higher sequencing depth usually increases the detection of less abundant viruses, and facilitates the assembly of more complete genomes [48,49].

## 4. Drylab Challenges in Describing the Virome Composition

In addition, the bioinformatic methods can affect the observed gut virome composition as well [48]. Here, we focus on three major challenges regarding (i) de novo assembly and the associated quality of viral assemblies [48], (ii) viral identification [50], and (iii) viral classification [51].

### 4.1. Challenges in Assembling Viral Genomes

First, de novo assemblers make use of algorithms capable of reconstructing large genomes from shorter reads, without a priori knowledge. They are pivotal in virome studies, as the majority of viral sequences do not share evident similarity to members of current reference databases [43], and therefore deem the usage of reference-based assemblers inadequate. Furthermore, Hill and colleagues observed that the type of assembler substantially affects the observed virome composition [48]. Here, the assemblers were shown to be affected by aspects inherent to the structure of virome data, such as an uneven or low sequencing coverage, and a high presence of repetitive regions within their genomes [52,53]. These difficulties lead to a lower quality of viral assemblies characterized by inaccuracy and a high percentage of genome fragmentation, thereby hampering compositional and diversity analysis [48,53]. For these reasons, the ‘Minimum Information about an Uncultivated Virus Genome (MIUViG)’ provides community-wide standards regarding the quality of viral genomes (e.g., completeness) and associated assembly software [54].

### 4.2. Challenges in Viral Identification

A number of viral identification tools have been developed to mine viral signals within de novo assembled metagenomic sequences [55,56,57,58,59,60,61]. Viral signals that cannot be detected by any of these tools are referred to as ‘viral dark matter’, and can vary greatly depending on the tool used [62]. Moreover, Garmaeva and colleagues performed a comparative analysis between viral identification tools and showed that no single tool can detect all viruses in every single dataset, but its performance rather depends on the specific type of data (e.g., sample type) [50]. Furthermore, the performance of most tools improved with a decreasing taxonomic complexity and an increasing sequence length [63]. Overall, large variations in the identifiable viral fraction, due to a high percentage of viral dark matter (>40%), create an inaccurate comparison of virome studies and limit our understanding of the virome complexity in health and disease. For example, Norman and colleagues observed for the first time that the gut virome in IBD patients was characterized by a high relative abundance of *Caudovirales* and a low *Caudovirales* richness, although only 15% of the sequences they generated were identified to be of viral origin [11]. Years later, Clooney and colleagues reanalyzed the dataset with distinct methods and found more than 80% of the metagenomic sequences to be of viral origin [64]. Simultaneously, they observed a higher phage richness in CD, but not UC patients, thereby highlighting the importance of viral dark matter to draw conclusions about the gut virome in IBD patients [64]. For those reasons, the community-wide MIUViG standards recommend the viral identification software as well [54]. In addition, virome studies often lack data regarding the identifiable viral fraction or the viral dark matter, which are essential for an accurate understanding of the virome in health and disease and complicate comparative analyses between virome studies.

### 4.3. Challenges in Viral Classification

The identifiable viral fraction is yet to be classified, which leads to the discussion of the last major challenge in describing the gut virome composition which is phage taxonomy. Phages have been historically classified, according to morphological features, into several taxonomic groups [65,66]. Consequently, the largest group of *Caudovirales* (e.g., tail dsDNA phages) is composed of three morphologically distinct groups, namely the families *Podoviridae*, *Siphoviridae*, and *Myoviridae* [67]. Recent works, based on viral genetic relatedness, repeatedly demonstrated a significant problem regarding paraphyly of these morphologically similar viruses [68,69,70]. This prompted the International Committee on Taxonomy of Viruses (ICTV) to shift from a morphology-based towards a genome-based phage taxonomy [51]. This shift is promoted by several efforts such as the introduction of the monophyletic subfamilies, and the abolishment of the *Caudovirales* order. The latter also coincided with the simultaneous lifting of their members to the *Caudoviricetes* class, in order to make room for new monophyletic orders [51,71,72]. In addition to those challenges, an inverse relationship exists between taxonomical resolution and the classifiable viral fraction, ascribed to the overwhelming amount of phages with low identity scores to the reference database [62]. The low similarities are insufficient for describing high resolution taxonomies (e.g., genus and species rank: 70% and 95% average nucleotide identity of entire genome), but are sufficient for low resolution taxonomies (e.g., class rank) [51,62,73]. Therefore, the higher the taxonomical resolution (e.g., genus rank), the lower the classifiable viral fraction. Reversely, the lower the taxonomical resolution (e.g., class rank), the higher the classifiable viral fraction. For that reason, virome studies have to carefully evaluate the needs and applications of their research before choosing the taxonomical rank to report on.

Taken together, it is clear that the implementation of distinct wetlab and drylab approaches substantially affect the reported gut virome composition. Regardless, to accurately describe the gut virome composition based on the previous literature, researchers should consider the methodological discrepancies between studies. To obtain a more accurate comparative analysis, the discussed studies should preferentially include information regarding the factors affecting the virome composition, such as VLP enrichment, random amplification, type of assembler, identifiable viral fraction, and the viral taxonomic rank.

## 5. The Virome Composition in a Healthy Human Gut

The largest known groups of viruses residing in a healthy human gut are members of the *Caudoviricetes* (tailed dsDNA) and *Malgrandaviricetes* (spherical ssDNA) phage classes [74,75]. Although little is known about the identity of their bacterial hosts, *Caudoviricetes* are thought to infect a diverse group of phyla, including *Firmicutes*, *Bacteroidetes*, *Actinobacteria*, *Proteobacteria*, and *Verrucomicrobia*, while *Malgrandaviricetes* are currently thought to infect either intracellular parasites (e.g., *Chlamydia*, *bdellovibrio*, and *Spiroplasma*) or *Enterobacteria* [76,77,78]. CrAss-like viruses are a recently identified monophyletic clade within the *Caudoviricetes* class, and are thought to be the most abundant viral group in the healthy human gut [75,79,80]. Shortly after the discovery of CrAss-like phages, several other highly abundant and prevalent viral clades were characterized including Gubaphage, LoVEphage, *Flandersviridae*, and giant Lak phages [78,81,82,83]. Viruses infecting eukaryotic cells, such as plant viruses, are also frequently observed in the healthy human gut, and can be present in high amounts [84]. They are likely obtained via the diet and from simply passing throughout the gastrointestinal tract [75,85]. Other frequently observed viral clades are the small circular single-stranded DNA viruses, including *Anelloviridae* and *Cressdnaviricota* [75,86]. Although the biological role of these viruses in the human body remains unclear, recent studies using new computational workflows have been able to predict several hosts such as *Entamoeba gingivali*, *Blastocystis* spp., and others [75,86,87]. Even though Anelloviruses are not very abundant, they represent an extremely diverse and prevalent (presumably) eukaryotic viral family in the early stages of life, seemingly declining when transitioning towards a mature microbiome [76]. In addition, Nishijima and colleagues studied the intestinal DNA virome in a population-based cohort of healthy individuals (*n* > 1000), and although they did not consider microviruses, they identified a large number of previously unknown viral clades [88]. Therefore, the description of the gut virome at the present time is likely only revealing a (small) part of the full viral complexity in the healthy human gut. 

This viral complexity is characterized by a remarkable individuality of viral contigs, or a high inter-personal variation [17,75]. Despite the large differences between individuals, the virome within an individual can be quite stable over time, as characterized by a low intra-individual variation [89,90]. Phages belonging to the lytic CrAss-like and *Malgrandaviricetes* groups have been detected in the human intestine for up to one year [17,75]. Interestingly, anelloviruses have even been detected in the human intestine for a period of more than one year, underscoring the stability of certain intestinal viruses over time [76]. Moreover, the mammalian gut virome revealed a distinct composition alongside its longitudinal axis (‘viral biogeography’) as well, indicating a higher diversity and viral load in the caecum and large bowel [91]. In general, the healthy gut microbiota is regarded as a diverse entity, in which the reduction of microbial (e.g., bacterial) diversity is oftentimes associated with human diseases [92]. More specifically, a common feature of irritable bowel syndrome, colorectal cancer, IBD, and other diseases is precisely this loss of microbial diversity [93,94,95]. Even though the diversity dynamics within the healthy gut virome are not fully understood yet, they are thought to be positively correlated with the aforementioned microbial diversity [96]. Moreover, Garmaeva and colleagues have shown that high viral diversity is associated with a more stable virome that is less responsive to external factors, yet reportedly fragmentation and bias may have affected their analysis [89]. Taken together, these results highlight that the healthy gut is composed of a relatively stable and specialized virota, in which a high diversity might stimulate resilience to change.

## 6. External Factors Shaping the Gut Virome Composition

Despite its capacity to exhibit resilience, even the most resilient virota will experience compositional changes over time [97]. The environmental factors, or covariates that lead to such changes, can be grouped into several major components such as anthropometric factors, diet and associated stool consistency, physical activity and lifestyle, medication, disease, and geography [75,88,98,99]. First of all, anthropometrics are measurements of features of the human body, such as body size, height, and many others. In this regard, age and Body Mass Index (BMI) are repeatedly shown as prominent anthropometric covariates of the gut virome [12,88,99,100]. Recently, multiple studies observed a positive association between age and viral diversity, in which age-dependent patterns revealed a high diversity in adulthood, followed by a lower diversity in the elderly [88,99]. Secondly, the dietary habits of individuals are believed to affect the viral composition in the gut as well [41,88,101]. Individuals with similar dietary habits revealed a more similar virome composition, which changed to a great extent with dietary intervention [41]. Moreover, a high-fat diet was characterized by a low relative abundance of *Caudoviricetes* and a high relative abundance of *Malgrandaviricetes* phages, and was accompanied by a lower lysogenic potential [101]. Other important dietary habits contributing to virome variation are dairy products, fruit intake, and coffee consumption [88,98]. The former two habits revealed a positive association with the Shannon diversity index, thereby once more indicating the importance of diet on viral dynamics [88]. In addition, the Bristol Stool Score (BSS), a measure to describe stool consistency, showed an important effect on the gut virome composition and is known to be linked with diet [88,102,103]. Third, physical activity and lifestyle (e.g., alcohol intake and smoking behavior) are recently identified covariates of the gut virome composition [88]. Even though not much is known about the specific effects on the virome, bacterial research suggests that exercise could be an important factor by promoting a high diversity and enhancing SCFA synthesis [104]. Next, medication and diseases are known covariates of the gut virome, as mentioned before [88,98,105]. Several studies showed that various medications have the potential to induce the activation of prophages within their respective host, thereby changing the composition of the gut virome [106,107]. Lastly, Zuo and colleagues investigated the fecal virome of 930 healthy individuals and found geographical location to be the largest explanatory factor, with a particular focus on rural and urban communities [98]. However, all the current descriptions of the healthy gut virome are subject to potential methodological biases, and remain to be validated in large cohorts of deeply phenotyped individuals. In addition, even the largest studies today can only partially explain the virome variation, emphasizing the large impact of stochastic and unknown deterministic processes shaping the virome composition, which are yet to be elucidated.

## 7. The Effect of External Factors on the Lifecycle of Phages

Regardless of their limited effect, particular external factors (e.g., medication and diseases) could introduce substantial changes by inducing prophages and thereby changing the phage lifecycle [64,106]. Therefore, it is thought that in a healthy state an intricate balance exists between lytic and lysogenic phages [108]. For example, higher microbial cell counts are associated with a lysogenic lifestyle, while lower microbial cell counts are associated with higher lytic activity [109]. Presumably, higher microbial densities offer phages a convenient way to reproduce their integrated genomes through bacterial replication, rather than lysing and killing the bacterial host [109]. Studies investigating the infant gut virome revealed a high relative abundance of *Caudoviricetes* phages (mostly lysogenic) at early stages, followed by a shift over time towards a high relative abundance of *Malgrandaviricetes* phages (obligatory lytic) [75,110]. This transitioning towards a more adult-like state is accompanied by a lower lysogenic, and a higher lytic potential of the gut virome [76]. In general, a healthy adult gut is marked by a complex balance between lysogenic and lytic phages, in which particular (external) factors can destabilize this intricate balance and further stimulate the induction of prophages.

## 8. Community Typing as a Novel Method for Analyzing the Gut Virome

The human gut virome is a complex entity and is composed of a large number of viral species [99,111]. This extreme diversity makes studying the virome quite a challenge, and further complicates the virome analysis. In bacterial research, the concept of community typing (‘enterotyping’) was introduced, as a means to reduce this complexity and collapse microbiome variation into a few categories [112]. Community typing employs the Dirichlet Multinomial Mixture (DMM) algorithm that is based on probabilistic modeling, and considers specific characteristics of microbiome data, such as sparsity [113]. Making use of this algorithm, samples of the same community (similar bacterial abundance profiles) are grouped into microbial configurations, without making any claims about the underlying discrete nature of the strata [113]. These methods consistently stratified the gut microbiota into four enterotypes, named *Bacteroides*1, *Bacteroides*2 (Bact2), *Prevotella*, and *Ruminococcus*, and revealed numerous associations with the aforementioned external factors, such as diet and diseases [114,115,116,117]. For example, the Bact2-enterotype is believed to reflect gut dysbiosis, having a high prevalence (80%) in IBD patients, and a rather low prevalence (15%) in the healthy population [114].

Given the extended knowledge enterotypes have provided in understanding the human gut microbiome [117,118,119], it has been speculated that a viral counterpart (‘viral community typing’) could be a useful tool to improve the understanding of the gut virota as well. In doing so, Song and colleagues investigated various published datasets and revealed that most individuals could be stratified based on their gut virota into two viral community types, although they were unable to describe their taxonomical composition due to a high degree of viral dark matter [120]. In addition, our group also previously investigated the gut virota in IBD patients, which confirmed the existence of two viral community types, and revealed that they were dominated by either CrAss-like and *Malgrandaviricetes* (‘community type CrM’), or *Caudoviricetes* phages (‘community type CA’) [121]. Furthermore, the latter viral community type was associated with a low virome diversity, the dysbiotic Bact2-enterotype, and active disease, thereby highlighting a clinical potential of such community types [121]. Even though it is still in its infancy, viral community typing is a promising future tool to explore viral compositional changes in health and disease.

## 9. Risk Factors Involved in the Development of IBD

The pathophysiology of IBD is unknown; however, the interplay between genetic susceptibility, the gut microbiota, immune responses, and environmental factors is suggested as a current working hypothesis [122]. Several prominent risk factors are involved in the development and/or progression of the disease and include genetics, smoking, diet, medication, mental health, stress, and many others [123,124,125,126,127,128,129]. The largest known risk factor is genetics, with up to 12% of patients revealing a family history of disease [124]. Moreover, twin studies showed a more prominent genetic impact in CD compared to UC pathophysiology [123]. Secondly, smoking was associated with a lower risk of developing UC and a higher risk of developing CD [125]. Another recognized risk factor contributing to disease is the patients dietary habits [126]. More specifically, a Western diet was previously linked to disease progression, and is characterized by low amounts of fruit and vegetables and high amounts of fat and proteins [126]. Bacterial fermentation of dietary fibers is known to produce large amounts of SCFA [127]. These metabolites provide a strong anti-inflammatory effect by preventing excessive immune response, and correspond to a reduced risk of developing IBD [128]. Besides diet, psychological stress and depression are associated with the development of IBD as well, and simultaneously with a weakening of the intestinal barrier and a disruption of the gut microbiota [129,130]. Taken together, these risk factors are implicated in the pathophysiology of IBD, either by increasing susceptibility to disease, acting as an environmental trigger, or stimulating disease progression. 

## 10. The Pathophysiology of IBD Is Associated with a Disrupted Gut Virome Composition

Simultaneously with disease development, the microbiota will shift towards an imbalanced microbial state. The gut virota is recognized to shift towards an imbalanced viral state as well, notwithstanding that studies revealed inconsistent results [11,64,79,131,132,133,134,135,136]. To accurately describe the consensus regarding the imbalanced IBD gut virome, methodological discrepancies between studies should be considered. As mentioned earlier, several methods are known to affect the virome composition, including VLP enrichment, random amplification, type of assembler, tools used to identify the viral fraction, viral taxonomic rank of the analyses, and are summarized in Appendix A for various IBD studies. Despite all the before-mentioned biases, the imbalanced IBD gut virome repeatedly revealed a higher relative abundance of *Caudovirales*, and a lower relative abundance of *Microviridae* phages [11,131,132,134,135] (Appendix A). This imbalanced viral state was further characterized by a low prevalence and abundance of CrAss-like phages, although one single study performed by Takayuki and colleagues reported a higher relative abundance of CrAss-like phages in CD patients [64,79,134] (Appendix A). The latter study was marked by low sample size (*n* < 20) and employed distinctive methods that performed viral identification/classification on read rather than contig level [134]. With regards to eukaryotic viruses, members of some viral families, such as *Anelloviridae* and *Herpesviridae*, revealed a higher prevalence in IBD patients compared to healthy controls [11,64,135] (Appendix A). However, these findings could not yet be reproduced and seem to be rather study-specific. Oftentimes virome studies do not incorporate a reverse transcription step, and as a result fail to capture the abundantly present RNA viruses in the eukaryotic virome. Finally, most studies characterizing the virome in IBD patients disregard the lifestyle of phages. Nevertheless, one recent study performed by Clooney and colleagues described a high lysogenic potential in CD, but not UC patients, suggesting a different role of lytic and lysogenic phages depending on the IBD subtype [64] (Appendix A).

## 11. Methodological Bias in Diversity Analysis Prevents Firm Conclusions

The viral imbalance in IBD patients can be described by diversity indices (e.g., observed richness and Shannon diversity) as well. In this respect, a high degree of fragmentation of viral genomes (due to insufficient sequence depth or assembler related issues) will automatically lead to a higher viral richness if no steps are taken to avoid this overestimation. Thus, to accurately describe diversity indices, the degree of fragmentation, or in other words, the quality of the assembled viral genomes should be considered. One of the ways is to only take the (near) complete phage genomes into account, as can be determined by tools, such as CheckV [137]. Currently, very few studies investigating the IBD virome employ stringent quality measures to obtain high-quality viral genomes, thereby questioning their reliability and complicating diversity comparisons between studies (Appendix A). Moreover, studies investigating fecal samples of UC patients did not reveal a distinct viral richness or diversity compared to healthy individuals [11,64]. After focusing on specific viral groups, *Microviridae* did not reveal a distinct richness; however, one study performed by Norman and colleagues did show a higher *Caudovirales* richness compared to healthy controls [11,64]. Importantly, the latter study incorporated a lenient 1 kb threshold for the length of viral genomes, apart from identifying only a minority of 15% of sequences as viral [11]. Reversely, Zuo and colleagues investigated mucosal samples of UC patients and found a lower phage diversity compared to healthy individuals. In addition, *Caudovirales* phages revealed a lower richness and diversity compared to healthy controls, while richness differences concerning *Microviridae* phages were not investigated [131]. With respect to CD patients, studies investigating faecal samples yielded inconsistent findings [11,64,134,135]. More precisely, some studies showed no differences in phage diversity [64,134] compared to healthy individuals, while others showed either a higher [11] or a lower phage diversity [135]. Nevertheless, after focusing on specific viral groups, *Caudovirales* phages consistently revealed a higher richness compared to healthy controls, and conversely *Microviridae* phages revealed either no distinct richness, or a lower richness [11,64]. It should also be noted that studies investigating mucosal samples in CD patients are scarce and often do not include diversity analysis [133,136]. 

Taken together, several risk factors are recognized in IBD and likely contribute to the emergence of an imbalanced gut virota. According to the literature, the main characteristics of this imbalanced gut virota include a contraction of CrAss-like and *Microviridae* phages, and simultaneously an expansion of *Caudovirales* phages. Finally, statements regarding richness and diversity may be considered premature, since most studies are highly biased (e.g., high fragmentation of viral genomes), yet initial reports point towards a higher *Caudovirales* richness in the fecal virome of CD patients.

## 12. The Inflammatory Positive Feedback Loop as a Working Hypothesis Underlying the Pathophysiology of IBD

To conclude this review, we introduce a concept called the “inflammatory positive feedback loop” and emphasize the role that viral imbalances may play in the pathophysiology of IBD (Figure 3 and Figure 4).

First of all, it is believed that an (unknown) environmental trigger causes an abnormal immune response against the gut microbiota in genetically predisposed individuals, leading to intestinal inflammation (Figure 3A) [122]. Along with smoking, stress, and medications, the Western diet is considered one of the major triggers [123,125,126]. The diet is generally high in sugar and fat and low in complex fiber [126]. It is believed to weaken the mucus layer that separates microbes from host cells under normal conditions [138]. More specifically, as a result of a low-fiber diet, the gut microbiota is believed to shift from degrading fiber-derived glycans to degrading mucus-derived glycans (e.g., Ruminococcus gnavus), damaging the protective layers of the gut and promoting inflammation and gut permeability [138,139]. This inflammatory state is characterized by a clear microbial imbalance, such as loss of bacterial (and possibly viral) diversity, contraction of CrAss-like and *Microviridae*, expansion of *Caudovirales*, and possibly more viruses with lysogenic capacity (Figure 3B) [11,64,79,131].

Second, a process called phage-mediated lysis is initiated, describing an inflammatory positive feedback loop between phage induction and intestinal inflammation (Figure 4) [140,141]. In this case, intestinal inflammation stimulates enterocytes to produce stressors (e.g., reactive nitrogen species and reactive oxygen species) which trigger a stress response from the host bacteria (“SOS response”) [142]. This response induces prophages to initiate the lytic life cycle, leading to lysis of the bacterial host cell [140,141]. The increased lysis of the bacterial host cell is accompanied by an increase in pathogen-associated molecular patterns (e.g., bacterial DNA, lipopolysaccharide) that further activate enterocyte receptors [140,141]. In turn, enterocytes produce more stressors and promote the induction of prophages, initiating a positive feedback loop and increasing the lytic activity in intestinal inflammation (Figure 4) [140,141]. 

After the onset of the inflammatory feedback loop, the immune system also becomes dysregulated, and changes from an immunoregulatory to an immunostimulatory state (Figure 3C). Here, the aforementioned breakdown of the mucus layer allows a larger amount of microbial antigens (e.g., LPS, viral antigens) to stimulate toll-like receptors on enterocytes and immune cells present in the epithelium [11,143,144,145,146]. Subsequently, this increase in inflammatory stimuli upregulates pro-inflammatory signaling pathways and downregulates anti-inflammatory signaling pathways [147,148]. Pro-inflammatory signaling pathways lead to increased production of ROS, NOS, interferon, and other pro-inflammatory cytokines [149,150,151]. These signaling pathways inhibit the proliferation of regulatory T cells and stimulate the proliferation of effector T cells, which in turn stimulate colitis. On the other hand, in a healthy gut, anti-inflammatory signaling pathways lead to a higher production of TGF-β, retinoic acid, and other anti-inflammatory cytokines [149,150]. These pathways stimulate the proliferation of regulatory T cells, and inhibit the proliferation of effector T cells, resulting in a stable, healthy gut (Figure 3) [152].

In summary, the inflammatory positive feedback loop is a (contributing) hypothesis underlying the pathophysiology of IBD. It is characterized by microbial (and especially viral) imbalance and a dysregulated immune system, underscoring the link between the microbiome and IBD.

**Figure 2 viruses-15-00173-f002:**
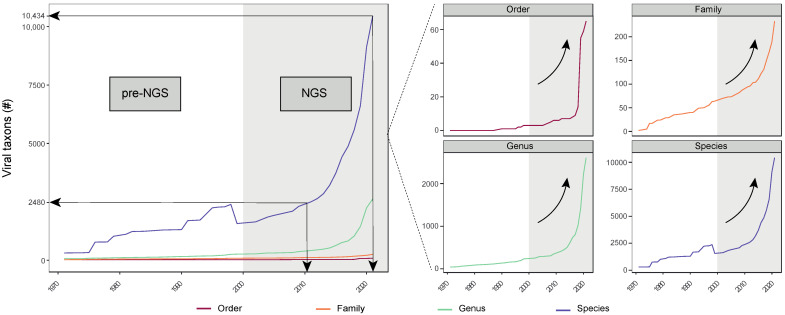
Expansion of viral taxa over time. Development of ICTV taxonomy across multiple taxonomical hierarchies (Order, Family, Genus, and Species) from the first public record in 1971 until 2021. The white and grey-colored area on the plot represents the viral taxonomies in the pre-NGS and NGS era, respectively. The arrival of NGS technology coincides with an exponential increase in viral taxa, especially in the last decade. For example, the number of viral species rose from 2480 to 10,434 between 2011 and 2021, which is more than a quadrupling of the known species in only one decade. Adapted from ref. [153].

**Figure 3 viruses-15-00173-f003:**
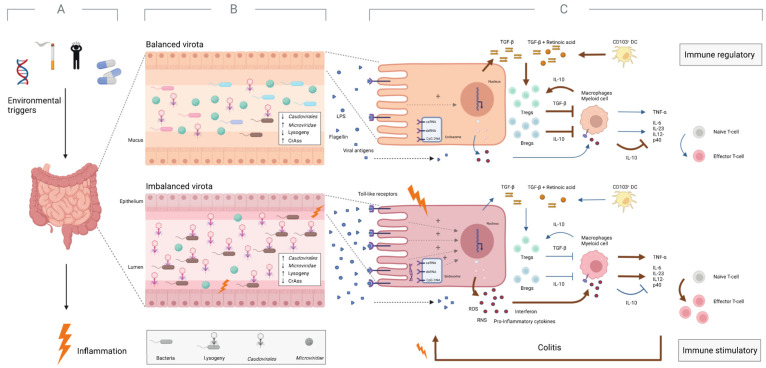
IBD patients possess an imbalanced intestinal virota and a dysregulated immune system under inflammatory conditions. (**A**) Environmental factors (e.g., smoking, stress, and Western diet) are believed to trigger intestinal inflammation in individuals predisposed to IBD. (**B**) This inflammatory condition is characterized by various viral imbalances compared to a healthy, non-inflamed gut. These imbalances include a high relative abundance of *Caudovirales* and a low relative abundance of *Microviridae*. In addition, the imbalances indicate a depletion of CrAss-like phages and possibly expansion of the lysogenic potential of the virome. (**C**) The altered intestinal virota is also associated with a dysregulated immune system. Here, breakdown of the mucus layer allows larger amounts of microbial antigens (e.g., viral antigens, flagellin) to stimulate Toll-like receptors in the intestinal mucosa, leading to the upregulation of proinflammatory signaling pathways and promotion of colitis. Red and blue arrows indicate the upregulated and downregulated pathways, respectively. Created with BioRender.com.

**Figure 4 viruses-15-00173-f004:**
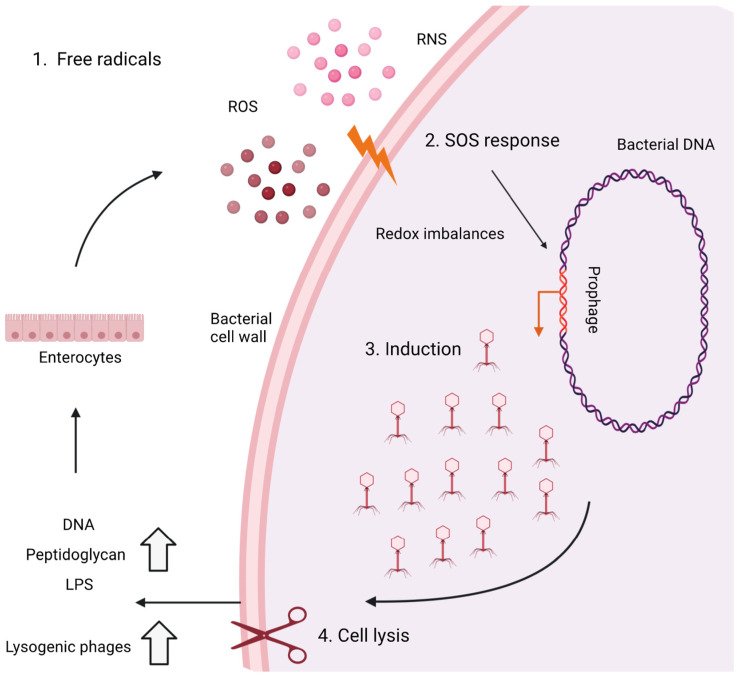
The hypothesis of phage-mediated lysis to sustain the positive inflammatory feedback loop in IBD patients. Intestinal inflammation stimulates the intestinal mucosa to produce stressors such as free radicals (e.g., ROS and NOS) (1) and triggers a stress response in the host bacteria called the “SOS response” (2). This response is a DNA damage response and leads to the induction of prophages (3), subsequent viral induction, and lysis of cells (4). Subsequently, pathogen-associated molecular patterns (e.g., LPS and peptidoglycan) are released, which in turn can stimulate the intestinal mucosa to produce further stressors, sustain the stress response in host bacteria, and thereby sustain the positive inflammatory feedback loop. Adapted from ref. [140] and created with BioRender.com.

## 13. Conclusions and Perspectives

Norman and colleagues first described viral alterations in the gut of IBD patients in 2015. Since then, numerous studies have described viral changes in the guts of these patients, but often with strong methodological biases that limit our ability to compare and reproduce these findings. Most notably, these limitations influenced the understanding of the presence of certain eukaryotic viruses, changes in viral richness and diversity, and changes in phage genera or family taxonomy. In this review, we highlight these methodological challenges that must be considered when describing the composition of the virome. Accordingly, it has been highlighted that when analyzing diversity, correcting for genome quality to avoid fragmentation bias appears to be of paramount importance. Another example is the attempt to work with sequence-based rather than morphology-based taxonomies. Considering these limitations, we nevertheless describe a consistent viral imbalance in IBD patients, characterized by a high relative abundance of *Caudovirales* and a low relative abundance of *Microviridae*. Other findings included a potentially higher lysogenic capacity and a lower prevalence of CrAss-like phages. In addition, the positive inflammatory feedback loop is suggested as a (contributing) hypothesis of sustained intestinal inflammation, and is associated with microbial (and viral) imbalances. In the end, we believe that this review highlights the challenges and changes regarding the gut virome, and paves the way for understanding the intrinsic role of viral imbalances in IBD.

## Figures and Tables

**Figure 1 viruses-15-00173-f001:**
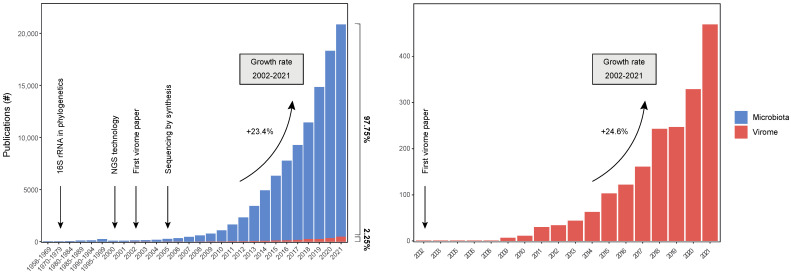
Number of publications in microbiome and virome research. Number of publications were determined by querying the search term “Microbiome [TITLE/ABSTRACT]” and “Virome [TITLE/ABSTRACT]” on PubMed. In the year 2000, the first next-generation sequencing technology (e.g., massive-parallel sequencing) was launched, along with the first microbiome publication on PubMed. The first virome publication was published in 2003 on PubMed. In the last decade (2010–2021), the microbiome and virome research showed an exponential increase in publications with an average growth rate of 33.4% and 28.1% per year, respectively. Nonetheless, in 2021, virome studies constituted only a small fraction (3.54%) of the total microbiome publications.

## Data Availability

Not applicable.

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
