# Peer review of "The Emerging Role of the Gut Virome in Health and Inflammatory Bowel Disease: Challenges, Covariates and a Viral Imbalance"

_viruses, 2023, doi:10.3390/v15010173_

Round 1

Reviewer 1 Report

The manuscript by Jansen and Matthijnssens present a critical view on what is known about the role of the gut virome in health and inflammatory bowel diseases (IBD), bringing out the various technical biases that prevail in these studies. Nevertheless, they highlight that phages affiliated to Caudovirales have higher richness in IBD than in healthy controls, across different studies (refs. 11, 43, 129). Finally, they present an hypothesis on the pathophysiology of IBD, where phage-bacteria interactions, and in particular prophage induction play a central role in generating an inflammatory positive feedback loop.

Major comments:

1.          In line 267 one can read: “”soft stool was associated with the lack of fruits and hard stool with both excessive fat intake and dairy consumption . I find these associations unexpected and I suggest either the authors provide additional references supporting them or remove the statement. As the ms states, there are technical and methodological bias, resulting in incorrect conclusions. The paper they cite, ref. 101, provides tables with nonsignificant associations between diet and stool consistency (Tables 2 and 3). The associations cited in this review, come from a machine learning approach. The authors themselves acknowledge limitations due to low sample size and unbalanced design.

2.          Phage taxa as Microviridae and Caudoviridae are used as basic units to compare diversity between IBD and healthy states. Given the high diversity of phages assigned to Caudovirales and their mosaic structure, could the authors find other relevant information? For example, taxa at higher resolution and host-range information.

3.          The hypothesis that is presented indicates the central role of prophages in inflammation onset and maintenance. Howwever, no link is made between the highest diversity of Caudovirales in IBD and this role for prophages. Is there data on the lifestyle of those increased Caudovirales viruses in IBD?

4.          How the IBD studies presented in the supplementary table were selected? In other words, keyword search in pubmed (what keywords?), filtering based on abstract or those are the only studies on impact of virome in IBD?

Other comments:

Figure 3, Figure 4: the figure quality is very low.

Line 67, the most abundant, not the largest

Line 177: demonstrated an significant : “a”, instead of “an”

as a results  -> as a result, line 425

Figure 1> there are panels A and B in the Figure, but no reference to the panels in the legend.

Figure 2 legend: “viral taxons ” -> viral taxa

Excel table: “protein db” among tools…, footnote mentions “viral nt db” , what is “RA” ? Add column with sample size.

Reviewer 2 Report

Jansen el al have written a review covering literature relating to the gut virome examining the current challenges within this field of research.

Comments:

  • Line 63 - Having a “Results” section in a review seems inappropriate, can this be removed?

  • Line 67 - “largest biological entity in the world”. This sentence is phrased poorly. Do authors mean biggest by physical size or by their abundance and diversity in the biosphere?

  • Line 73 – Inoviridae phages infect by a third approach (chronic infection) that does not result in lysis of the host. Is there much literature looking at these phages within the human gut?

  • Line 78 to 79 - A prophage is a viral genome that infects bacterial cells and integrates with the bacterial genome whilst a provirus is a viral genome which integrates into a eukaryotic genome. - please correct.

  • Line 94 to 95 – beneficial/ detrimental to the bacterial host or human host in which the bacteria resides? Please clarify.

  • Line 99 to 100 – remove “hereby creating a pathogenic environment in the human gut” – this statement doesn’t make sense.

  • Line 105 - “drylab (bioinformatic) “ consider using the term in-silico.

  • Line 114 – what are the differences between the purification steps of centrifugation and ultracentrifugation?

  • Line 122 – Is there any literature that has looked into long (i.e. nanopore) and short (i.e.) sequencing approaches for sequencing of the gut virome?

  • Line 390 to 394 – Was the same observation made on reanalysis done by the Hill group for the Norman dataset? What cut-off points were used for viral contig length by the latter group and did these cut-off point impact observations made compared to the original study?

  • Line 424 to 426 – Might be nice to make a quick comment on how this shift into mucin-degrading glycans may be related to the increased prevalence of certain gut bacteria species. I.e. Ruminococcus gnavus.
